# Greener Synthesis of Pristane by Flow Dehydrative Hydrogenation of Allylic Alcohol Using a Packed-Bed Reactor Charged by Pd/C as a Single Catalyst

**DOI:** 10.3390/molecules26195845

**Published:** 2021-09-27

**Authors:** Takayoshi Kasakado, Yuki Hirobe, Akihiro Furuta, Mamoru Hyodo, Takahide Fukuyama, Ilhyong Ryu

**Affiliations:** 1Organization for Research Promotion, Osaka Prefecture University, Sakai, Osaka 599-8531, Japan; t_kasakado@c.s.osakafu-u.ac.jp (T.K.); hyodo_mam@c.s.osakafu-u.ac.jp (M.H.); 2Department of Chemistry, Osaka Prefecture University, Sakai, Osaka 599-8531, Japan; ykhirobe@gmail.com (Y.H.); akihirofuruta5@gmail.com (A.F.); 3Department of Applied Chemistry, National Yang Ming Chiao Tung University (NYCU), Hsinchu 30010, Taiwan

**Keywords:** flow dehydration, flow hydrogenation, Pd on carbon, dual-function catalyst, pristane

## Abstract

Our previous work established a continuous-flow synthesis of pristane, which is a saturated branched alkane obtained from a Basking Shark. The dehydration of an allylic alcohol that is the key to a tetraene was carried out using a packed-bed reactor charged by an acid–silica catalyst (HO-SAS) and flow hydrogenation using molecular hydrogen via a Pd/C catalyst followed. The present work relies on the additional propensity of Pd/C to serve as an acid catalyst, which allows us to perform a flow synthesis of pristane from the aforementioned key allylic alcohol in the presence of molecular hydrogen using Pd/C as a single catalyst, which is applied to both dehydration and hydrogenation. The present one-column-two-reaction-flow system could eliminate the use of an acid catalyst such as HO-SAS and lead to a significant simplification of the production process.

## 1. Introduction

Pristane, 2,6,10,14-tetramethylpentadecane **3**, is a saturated branched alkane obtained from Basking Sharks [1,2], and is known to induce autoimmune diseases in rodents. Pristane, however, is now being widely used as an adjuvant for monoclonal antibody production [3,4,5]. In 2007, the Fukase group reported the flow/batch combined synthesis of pristane **3** via a two-step procedure with allylic alcohol **1** as a key component. Flow dehydration was performed by using a stoichiometric p-TsOH. The resultant tetraene **2** can be reduced to pristane **3** in a batch flask using molecular hydrogen with Pd on carbon (Pd/C) as the catalyst [6]. To realize a greener process, the flow dehydration protocol that employs a stoichiometric amount of *p*-TsOH must be avoided. Then, we focused on silica-supported sulfonic acids (SAS) [7,8], for the flow dehydration step. Consequently, we found that the use of hydroxy-functionalized sulfonic acid silica, HO-SAS [9,10,11], for flow-dehydration worked well. Combined with the flow hydrogenation using Pd/C, we completed an acid waste-free flow synthesis of pristane (Figure 1) [10]. Since Pd/C often causes dehydrative hydrogenation of allylic alcohols [12,13,14,15,16,17], we were curious as to whether pristane **3** could be synthesized using only a Pd/C catalyst. In this communication, we are pleased to report a simple protocol for the flow synthesis of pristane using Pd/C as a single catalyst in a dual function, which avoids the need to use an acid catalyst. We believe that the present flow protocol would be useful in designing a simplified flow-production process of pristane. It should be stated that, although flow hydrogenations of a variety of organic compounds have widely been developed [18,19,20,21,22,23,24,25], there were no reports on flow dehydrative hydrogenation of allylic alcohols before this work.

## 2. Results and Discussion

We started with a batch reaction of triene-type alcohol **1**. As expected, when the reaction of **1** (0.5 mmol) was carried out in the presence of 10% Pd/C (20 mg, 0.02 mmol of Pd) in ethyl acetate as a solvent under atmospheric pressure of H_2_ at room temperature for 16 h, the desired pristane **3** was obtained in a 30% yield, together with saturated alcohol **4** (48%) (Table 1, entry 1). When we used Pd/Al_2_O_3_ instead of Pd/C, the reaction gave alcohol **4** in a 92% yield along with a trace amount of pristane **3** (Table 1, entry 2). These results strongly suggest that the Pd/C catalyst has an exceptional ability to reduce an allyl alcohol moiety to pristane **3**, which was consistent with results reported by other groups [10,11,12,13,14,15]. Under harsh conditions that included higher temperature (90 °C) and higher pressure (10 atm), the yield of **4** was increased, but that of **3** did not change (Table 1, entry 3).

Despite the low yield, the results of the batch reaction confirmed that pristane **3** was formed from **1** via the use of a single catalyst, Pd/C. Nevertheless, the comparable selectivity of **3**/**4** obtained by a batch reaction was rather disappointing. Next, we examined the flow dehydration/hydrogenation sequence, which improved the ratio of **3**/**4** dramatically (Figure 2). Thus, when an AcOEt solution of **1** was mixed with H_2_ (13 atm) using a static mixer (500 µm i.d.) followed by passage through a column reactor (4 mm i.d., 15 cm length) packed with Pd/C (575 mg, 0.54 mmol of Pd) at 90 °C, to our surprise, the reaction gave pristane **3** as the major product in a 59% yield along with an 18% yield of alcohol **4** with only a 30 sec residence time. Heating was important, since the flow reaction at room temperature gave low yields of **3** (15%) and **4** (3%). We speculated that the use of a packed-column flow reactor ensured larger catalyst/substrate ratios which brought a high contact frequency between the substrate and catalyst to cause the efficient dehydrative hydrogenation of an allyl alcohol moiety to give pristane **3**.

The flow procedure resulted in excellent selectivity over the batch reaction. Then, we decided to use model compound **5** to see if the product selectivity could be further improved (Figure 3). When the reaction of **5** (0.5 mmol) was carried out in the presence of 10% Pd/C (20 mg) in AcOEt under atmospheric H_2_ at room temperature for 24 h, the desired alkane **6** was obtained as a major product, together with hydrogenated alcohol **7** and ketone **8** in a ratio of **6**/**7**/**8** = 64/16/20 (by a GC analysis). Interestingly, the addition of a 10% volume of *i*-PrOH to AcOEt improved the formation ratio of alkane **6** (**6**/**7**/**8** = 78/10/12). Since ketone **8** was also detected, we, then, carried out time course experiments. Interestingly, the reaction in AcOEt as a sole solvent did not proceed for the first 30 min. After 1 h, around 20% of alcohol **7** was formed, and, then, alkane **6** was rapidly formed (Figure 3a). On the other hand, the reaction with EtOAc/i-PrOH was started within 30 min and ended in 1 h, which gave a higher yield of **6** (Figure 3b). These results suggest that the reaction had an induction period and the addition of i-PrOH and initially formed secondary alcohol **7** contributed to generate H-Pd species prior to the reaction of molecular hydrogen with Pd. Since ketone **8** was formed after alcohol **7** was formed, the transfer hydrogenation from the secondary alcohol **7** to **8** [26,27,28] was more likely than olefin-isomerization of the allylic alcohol **5** [29,30,31,32].

It seems quite difficult to speculate the mechanism of Pd-catalysis; however, one possible mechanism is shown in Figure 4. The transfer hydrogenation would produce a Pd-H species. Allylic alcohol **5** coordinated to the Pd catalyst in both the C–C double bond and hydroxy group. Then, the dehydration reaction proceeded to give a diene [33,34]. Finally, the diene was hydrogenated by Pd/C to give alkane **6**. It should be noted that the addition of Et_3_N inhibited the present dehydrated hydrogenation and only hydrogenated alcohol **7** was formed in 96% yield. This suggests that a base such as Et_3_N deactivated the acid functionality of Pd/C.

Since the addition of *i*-PrOH improved the product selectivity, we, finally, examined the flow synthesis of pristane **3** using a mixed solvent of AcOEt and *i*-PrOH (10 vol%) under continuous flow conditions (Figure 5). To our delight, the flow conditions worked far better to give pristane **3** in an improved yield of 71% with a reduced yield of alcohol **4** (10%), whose result was in good agreement with the model reaction.

## 3. Materials and Methods

### 3.1. General

The ^1^H NMR spectra were recorded using JEOL ECS-400 (400 MHz) spectrometers in CDCl_3_ and were referenced at 7.26 ppm for CHCl_3_. The ^13^C NMR spectra were recorded using JEOL ECS-400 (100 MHz) spectrometers in CDCl_3_ referenced at 77.0 ppm. Chemical shifts were reported in parts per million (δ). Splitting patterns were indicated as follows: br, broad; s, singlet; d, doublet; t, triplet;, m, multiplet. GC analysis was performed on a Shimadzu GC-2014 instrument equipped with an FID detector using a J&W Scientific (Hongkong, China) DB-1 column under the following conditions: initial oven temperature was held at 60 °C for 5 min, the first ramp was 20 °C/min to 250 °C, which was held for 5 min. The products were purified by flash column chromatography on silica gel (KANTO CHEMICAL CO., INC., Tokyo, Japan, Silica Gel 60N (spherical, neutral, 40–50 μm)). HRMS spectra were recorded on BRUKER micrOTOF-II. Allylic alcohol **1** was prepared according to a previously established procedure found in the literature [6]. We purchased 10% Pd/C from Sigma-Aldrich Co. LLC. (St. Louis, MO, USA). (10% Pd on carbon, average particle size of 15 μm) and used it as received. Stainless steel columns (4.0 mm i.d. × 50 mm or 150 mm) were purchased from GL Sciences Inc (Tokyo, Japan). The column reactor and microtube were connected with PEEK fittings (GL Sciences Inc., 1/16′′). The micromixer (500 µm i.d.) was purchased from GL Sciences. A back-pressure regulator was purchased from DFC Inc. (Woodland, CA, USA). The solution was introduced into the flow microreactor system using an HPLC pump, PU714 (GL Sciences Inc.).

### 3.2. Typical Procedure for the Batch Synthesis of 2,6-Dimethylheptane ***6***

To a 10 mL glass flask, 2,6-dimethyl-2-hepten-4-ol **5** (0.5 mmol, 71 mg) and Pd/C (20 mg) were added along with a solvent (AcOEt, 3 mL). The mixture was stirred at room temperature under H_2_ (1 atm). After the reaction, the reaction mixture was filtered to remove Pd/C and an aliquot of the solution was applied to GC analysis.

### 3.3. Typical Procedure for the Batch Synthesis of Pristane ***3***

Allylic alcohol **1** (0.5 mmol, 139 mg), Pd/C (20 mg) and EtOAc (3 mL) were placed in a 10 mL glass flask. The mixture was stirred at room temperature for 16 h under H_2_ (1 atm). After the reaction, the solvent was evaporated and the residue was filtered through a silica-gel pad and washed with *n*-hexane to give pristane (**3**) (39.7 mg, yield 30%, purity 100%). The silica-gel pad was washed with EtOAc to give **4** (68.8 mg, yield 48%).

### 3.4. Procedure for the Flow Synthesis of Pristane ***3***

Allylic alcohol **1** (477 mg) was dissolved in EtOAc (10 mL) and pumped using an HPLC pump (flow rate = 0.5 mL/min). The H_2_ gas (flow rate = 2.6 mL/min) was supplied through a mass flow controller. Both the solution of **1** and hydrogen gas were introduced to a T-shaped mixer (i.d. 500 µm), and the resultant mixture was then introduced into a stainless steel column (inner volume: 1.44 mL, 4.0 mm i.d. × 150 mm) filled with a Pd/C catalyst (595 mg) with an outlet that was connected to a back-pressure regulator (1.3 MPa). The solution eluted during the first 10 min was discarded. The remaining solution was collected for 6 min in a glass flask, and the solvent was evaporated. The crude product was applied to silica-gel column chromatography and the fractions eluted with *n*-hexane gave pristane (**3**) with inseparable by-products (96 mg, yield 71%, purity 93% (determined by GC)). The fraction eluted with ethyl acetate gave saturated alcohol **4** (14 mg, yield 10%).

2,6,10,14-Tetramethylpentadecane (pristane, **3**) (see Appendix A)

Colorless oil; ^1^H NMR (CDCl_3,_ 400 MHz) δ 1.58–1.44 (m, 2H), 1.40–1.00 (m, 20H), 0.91–0.81 (m, 18H); ^13^C NMR (CDCl_3_, 100 MHz) δ 38.7, 37.7, 33.1, 28.2, 25.1, 24.8, 23.0, 20.0.

2,6,10,14-Tetramethylpentadecan-4-ol (**4**) (see Appendix A)

Colorless oil; ^1^H NMR (CDCl_3,_ 400 MHz) δ 3.83–3.70 (m, 1H), 1.83–1.70 (m, 1H), 1.70–1.00 (m, 20H), 0.80-1.00 (m, 18H); ^13^C NMR (CDCl_3_, 100 MHz) δ 68.1, 67.7, 47.7, 47.2, 46.0, 45.9, 45.7, 45.6, 39.4, 38.3, 37.4, 37.3, 37.3, 37.1, 37.0, 32.9, 29.7, 29.6, 29.3, 28.1, 24.9, 24.7, 24.6, 24.3, 23.7, 23.5, 22.8, 22.7, 22.3, 20.5, 20.4, 19.8, 19.7, 19.4. HRMS: *m/z* calcd for C_19_H_40_NaO (M^+^+Na) 307.2977, found 307.2952.

## 4. Conclusions

We showed that, in the presence of dihydrogen and a catalytic amount of Pd/C, the dehydrative hydrogenation of triene-alcohol **1** proceeded, in a batch flask, to give the desired pristane **3** and saturated alcohol **4** in a nearly comparable ratio. Interestingly, when continuous flow conditions were applied, a dramatic preference of **3** over **4** was observed by as much as 7/1. Larger catalyst/substrate ratios, ensured by the use of a packed-column reactor, were likely to cause the efficient dehydrative hydrogenation of an allyl alcohol moiety by reacting with more abundant active Pd-H species. The flow reaction proceeded quickly at 90 °C and needed only 30 s of residence time, which established the efficacy of the consecutive flow reaction for production compared with a batch reaction. Such a strong boost by a flow system is a novel result, which could, undoubtedly, lead to a greener and inexpensive production of pristane and some other target compounds. This one-column-two-reaction-flow system could eliminate the use of an acid catalyst such as HO-SAS, which also leads to a significant simplification of the facile production process for dehydrative hydrogenation of alkenyl alcohols.

## Data Availability

Not applicable.

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
