# Peer review of "Greener Synthesis of Pristane by Flow Dehydrative Hydrogenation of Allylic Alcohol Using a Packed-Bed Reactor Charged by Pd/C as a Single Catalyst"

_molecules, 2021, doi:10.3390/molecules26195845_

Round 1

Reviewer 1 Report

This work focuses on the greener synthesis of pristane by Flow-Dehydration-Hydrogenation of allylic alcohol using a packed-bed eeactor xharged by Pd/C. The innovation of the article is sufficient, and it also has strong industrialization significance. This manuscript could be published after addressing the following questions:

  1. The information about the Pd/C catalyst, including loading amount, Pt particle size, specific surface area, etc., needs to be supplemented in this work.
  2. Why can the introduction of isopropanol improve the selectivity of this reaction? The author should give some explanation. In addition, what effect will temperature have on this reaction?
  3. The author should briefly discuss the reaction mechanism of EtOAc to pristane over Pd/C catalyst. Reference [Applied Catalysis B: Environmental 284 (2021) 119803; ACS Catalysis, 2021, 11, 6371−6383] can be referenced to this discussion.
  4. The author developed an acidic catalyst in their previous work. In this work, Pd/C was used as the catalyst. Although the cost of the catalyst was increased, the yield of the target product was greatly improved (yield > 90%). Does it have certain advantages from the perspective of industrial economy? Green Chem., 2021, 23, 3664-3676 can be used for reference to discuss economic issues.

Author Response

Reviewer 1

This work focuses on the greener synthesis of pristane by Flow-Dehydration-Hydrogenation of allylic alcohol using a packed-bed reactor charged by Pd/C. The innovation of the article is sufficient, and it also has strong industrialization significance. This manuscript could be published after addressing the following questions:

1. The information about the Pd/C catalyst, including loading amount, Pt particle size, specific surface area, etc., needs to be supplemented in this work.

  • We added the information of the Pd/C catalyst (catalyst loading, particle size, and provided source).

2. Why can the introduction of isopropanol improve the selectivity of this reaction? The author should give some explanation. In addition, what effect will temperature have on this reaction?

  • We added a result for a time-course experiment without i-PrOH and compared it with a time course experiment with i-PrOH. It was found that the addition of i-PrOH accelerated the reaction.  While the exact role of the i-PrOH for reactivity and selectivity is still unclear, i-PrOH may act as a Lewis base to accelerate the dehydration step.  We rewrote some sentences surrounding Scheme 3.

Interestingly, the reaction in AcOEt as a sole solvent did not proceed for the first 30 min. After 1 h, around 20% of alcohol 7 was formed, and then the alkane 6 was rapidly formed (Scheme 3(a)).  On the other hand, the reaction with EtOAc/i-PrOH was started within 30 min and ended in 1 h, which gave higher yield of 6 (Scheme 3(b)).  These results suggest that the reaction has an induction period and the addition of i-PrOH and initially formed secondary alcohol 7 contributes to generate H-Pd species prior to the reaction of molecular hydrogen with Pd.  Since the ketone 8 was formed after the alcohol 7 was formed, the transfer hydrogenation from the secondary alcohol 7 to 8 [26-28] is more likely than olefin-isomerization of the allylic alcohol 5.[29-32] 

  • We consider that the effect of temperature is simply the acceleration of the reaction. For comparison, we added the result for flow reaction at room temperature, which gave only a low yield of pristane. We added the following sentences before Scheme 2.

Heating is important, since the flow reaction at room temperature gave low yields of 3 (15%) and 4 (3%).  We speculate that the use of a packed-column flow reactor ensures larger catalyst/substrate ratios which brought high contact frequency between substrate and catalyst to cause the efficient dehydrative hydrogenation of an allyl alcohol moiety to give pristane 3.

3. The author should briefly discuss the reaction mechanism of EtOAc to pristane over Pd/C catalyst. Reference [Applied Catalysis B: Environmental 284 (2021) 119803; ACS Catalysis, 2021, 11, 6371−6383] can be referenced to this discussion.

  • Presumably, the reviewer makes misunderstanding. EtOAc is not involved in the reaction.  We added a possible reaction mechanism as a new Scheme 4 via dehydrative hydrogenation of allylic alcohol to alkane.  We carefully read these two papers suggested by the reviewer, which deal with detailed mechanism for Mn2O3-catalyzed oxidation of ethylene glycol and glycerol.  However, we could not find any direct relevance to our present manuscript. Therefore, we did not cite these references. We added the following sentences before new Scheme 4.

It seems quite difficult to speculate the mechanism of Pd-catalysis, however, one possible mechanism is shown in Scheme 4.  The transfer hydrogenation would produce Pd-H species.  Allylic alcohol 5 coordinates to Pd catalyst in both C-C double bond and a hydroxy group.  Then, the dehydration reaction proceeds to give a diene.[33,34]  Finally, the diene was hydrogenated by Pd/C to give the alkane 6.  It should be noted that the addition of Et3N inhibited the present dehydrated hydrogenation and only hydrogenated alcohol 7 was formed (eq. 1).  This suggests that a base such as Et3N deactivates the acid functionality of Pd/C.

4. The author developed an acidic catalyst in their previous work. In this work, Pd/C was used as the catalyst. Although the cost of the catalyst was increased, the yield of the target product was greatly improved (yield > 90%). Does it have certain advantages from the perspective of industrial economy? Green Chem., 2021, 23, 3664-3676 can be used for reference to discuss economic issues.

  • The cost of the catalysts is decreased compared with our previous study, which used both acid catalyst (HO-SAS) and Pd/C catalyst, since the present method does not require the use of acid catalyst. In addition, the current one-column-two-reaction-flow system would lead to a significant simplification of the production process. We added a brief explanation of these points in the conclusion part.

Reviewer 2 Report

attached 

Author Response

Reviewer 2

Abstract

The abstract is too short. I suggest the authors remove these beginning sentences: “Our previous work established a continuous-flow synthesis of pristane, …. Flow hydrogenation using molecular hydrogen via a Pd/C catalyst followed.”. The abstract should include a one sentence purpose followed by main results and conclusion.

  • We have added the following sentence in the abstract to clarify the significance of this work.

 The present one-column-two-reaction-flow system could eliminate the use of an acid catalyst such as HO-SAS and lead to a significant simplification of the production process.

Introduction

  1. It is too short
  2. Table 1 and the sentences on previous work may be removed and replaced by a sentence along with the references. More references may be added on work done by other researchers. Then add why the topic is important, what has been researched already and what new is reported in this article.
  3. Need complete rewriting.
  • We added some sentences and references regarding SAS-dehydration and flow hydrogenation, which would help readers understand the background better. The reaction system also simplifies the production process, and we added explanation about this point.  We believe that Scheme 1 (“Table 1” in reviewer’s comment may be “Scheme 1”) and the sentences on previous work are necessary for readers to understand the background of this work. We rewrote the introduction part as follows.

Pristane, 2,6,10,14-tetramethylpentadecane 3, is a saturated branched alkane obtained from Basking Sharks,[1,2] and is known to induce autoimmune diseases in rodents.  Pristane, however, is now being widely used as an adjuvant for monoclonal antibody production.[3-5]  In 2007, the Fukase group reported the flow/batch combined synthesis of pristane 3 via a two-step procedure with allylic alcohol 1 as a key component.  Flow dehydration was performed by using a stoichiometric p-TsOH.  The resultant tetraene 2 can be reduced to pristane 3 in a batch flask using molecular hydrogen with Pd on carbon (Pd/C) as the catalyst.[6]  To realize a greener process, the flow dehydration protocol that employs a stoichiometric amount of p-TsOH must be avoided.  Then, we focused on silica-supported sulfonic acids (SAS),[7,8] for the flow dehydration step.  Consequently we found the use of hydroxy-functionalized sulfonic acid silica, HO-SAS,[9-11] for flow-dehydration worked well.  Combined with the flow hydrogenation using Pd/C, we completed acid-waste-free flow synthesis of pristane (Scheme 1).[10]  Since Pd/C often causes dehydrative hydrogenation of allylic alcohols,[12-17] we were curious as to whether pristane 3 could be synthesized using only a Pd/C catalyst.  In this communication, we are pleased to report a simple protocol for the flow synthesis of pristane using Pd/C as a single catalyst in a dual function, which would avoid the need to use an acid catalyst.  We believe that the present flow protocol would be useful in designing a simplified flow-production process of pristane.  It should be stated that although flow hydrogenations of a variety of organic compounds have widely been developed,[18-25] there are no reports on flow dehydrative hydrogenation of allylic alcohols before this work.  

  Results and Discussion

  1. I do not see any discussion of results. The results are not explained properly.
  2. Why NMR was done, no data provided, and nothing is explained.
  3. How the catalyst was synthesized/obtained, catalyst characterization??
  4. Add more references in this section.
  • We added some new results (Scheme 3(a), eq 1) and a proposed mechanism (Scheme 4) and the explanations for the formation of products.
  • We used NMR to confirm the structure of products, and spectrum data were provided in the experimental section. We provided1H-NMR and 13C-NMR charts as supporting information.
  • Pd/C was purchased from Sigma-Aldrich Co. LLC. and used as it received. Since particle size was missing in the original manuscript, we added the particle size of Pd/C in the experimental section.
  • According to the addition of the formation of products, we added some references on Pd-catalyzed transfer hydrogenation, alkene isomerization, and dehydration.